# Metabolomics in Central Sensitivity Syndromes

**DOI:** 10.3390/metabo10040164

**Published:** 2020-04-24

**Authors:** Joseph S. Miller, Luis Rodriguez-Saona, Kevin V. Hackshaw

**Affiliations:** 1Department of Medicine, Ohio University Heritage College of Osteopathic Medicine, Dublin, OH 43016, USA; Mille718@miamioh.edu; 2Department of Food Science and Technology, Ohio State University, Columbus, OH 43210, USA; rodriguez-saona.1@osu.edu; 3Department of Internal Medicine, Division of Rheumatology, Dell Medical School, The University of Texas, 1701 Trinity St, Austin, TX 78712, USA

**Keywords:** metabolomics, central sensitivity, fibromyalgia, biomarker, laboratory diagnostics, functional syndrome, overlapping syndromes, chronic pain, metabolite profiling, pain

## Abstract

Central sensitization syndromes are a collection of frequently painful disorders that contribute to decreased quality of life and increased risk of opiate abuse. Although these disorders cause significant morbidity, they frequently lack reliable diagnostic tests. As such, technologies that can identify key moieties in central sensitization disorders may contribute to the identification of novel therapeutic targets and more precise treatment options. The analysis of small molecules in biological samples through metabolomics has improved greatly and may be the technology needed to identify key moieties in difficult to diagnose diseases. In this review, we discuss the current state of metabolomics as it relates to central sensitization disorders. From initial literature review until Feb 2020, PubMed, Embase, and Scopus were searched for applicable studies. We included cohort studies, case series, and interventional studies of both adults and children affected by central sensitivity syndromes. The majority of metabolomic studies addressing a CSS found significantly altered metabolites that allowed for differentiation of CSS patients from healthy controls. Therefore, the published literature overwhelmingly supports the use of metabolomics in CSS. Further research into these altered metabolites and their respective metabolic pathways may provide more reliable and effective therapeutics for these syndromes.

## 1. Potential of Metabolomics to Identify Biomarkers for Central Sensitivity Syndromes (CSS)

Metabolomics involves the identification, quantification, and analysis of copious small molecules in a biological sample [1]. Sir Archibald Garrod provided one of the earliest accounts of this field when he described “inborn errors of metabolism” as possible triggers in disease processes [2]. Methodology for metabolite analysis has developed considerably over the last century. Visual inspection (alkaptonuria) [2], enzymatic detection of plasma metabolites, thin layer chromatography (TLC), gas chromatography (GC), and high-performance liquid chromatography (HPLC) can all help distinguish metabolites of similar classes including fatty acids, organic acids, amino acids and steroid alcohols [3]. However, recent years advancements in high-throughput metabolite profiling instrumentation, data analysis software, and databases have allowed the development of untargeted metabolomic approaches by which a broad range of metabolites, known and novel structures yet to be elucidated, are surveyed to identify novel biomarkers for disease diagnosis, prognosis and help to elucidate etiopathogenesis. Analytical platforms best suited for metabolomics studies employ hybrid systems based on separation methods (ultra-performance liquid chromatography (UPLC) or gas chromatography) in tandem with mass spectrometry (MS) and nuclear magnetic resonance spectroscopy (NMR) detection techniques, providing unique selectivity and sensitivity for metabolite profiling [3]. Tandem MS (MS/MS) involving fragmentation of analytes between stages increases specificity and allows for the use of MS without prior separation [4]. Vibrational spectroscopic methods such as Fourier transform-infrared spectroscopy (FT-IR) and Raman Spectroscopy have emerged as powerful fingerprinting techniques in metabolomics [5]. These techniques permit rapid, high-throughput, non-destructive analysis of biological samples and the resulting spectra allow for characteristic fingerprints of substances such as lipids, proteins, nucleic acids, polysaccharides, and phosphate-carrying compounds in the sample. Combining these laboratory techniques with pattern recognition statistical approaches, such as principal component analysis (PCA), soft independent modeling of class analogy (SIMCA), partial least squares discriminant analysis, artificial neural networks (ANNs), and support vector machines (SVM), enables the development predictive algorithms capable of distinguishing diseased from healthy subjects.

Untargeted metabolomics (Figure 1) searches for metabolites associated with biological pathways that contribute to disease pathogenesis and allows for the identification of molecular mechanisms in disease onset and development [6,7]. The goal of biomarker development in metabolomics is to create a predictive model from a collection of compounds that are used to classify new samples/persons into specific groups (i.e. healthy vs. diseased) with optimal sensitivity and specificity. Machine learning techniques in biomarker development are employed to identify an optimal subset of features that provide maximal discriminating power, to assess and validate the panel of biomarkers, and to develop an algorithm to predict a particular clinical outcome [8]. 

While the interest in metabolomic biomarkers has been growing exponentially for chronic diseases, the number of metabolomics-based for central sensitivity syndromes (CSS) is limited because of the challenges in accurately diagnosing these conditions. Central sensitization syndromes (CSS) are a collection of frequently painful disorders that generally lack a reliable diagnostic test [9]. Many of these conditions make up a compilation of syndromes called chronic overlapping pain conditions. The overlap amongst these conditions coupled with the co-occurrence of various psychiatric disorders likely reflects shared neurotransmitters in the underlying pathophysiology’s. Data suggest that affected individuals exhibit aberrant modulation of pain or sensory amplification rather than a structural or inflammatory condition in the specific body region where the pain is being experienced [9]. In addition to altered pain thresholds, responsiveness to auditory or visual stimuli may be similarly heightened [10,11]. 

Our initial understanding of pain processing in the Central Nervous System (CNS) thought of it as a complete circuit conveying input from sensory afferents to cortical pain processing centers. Melzack and Wall identified modulations to this circuitry by identification of inhibitory controls [12]. The discovery that certain neurons could reduce their thresholds following injury provided early evidence for sensitization [13,14,15]. Subsequent findings that contributed to our current understanding of sensitization included the discovery of the concept called windup in dorsal horn neurons, characterized by repeated stimulation of a nerve leading to progressive increase in action potentials over the course of the stimulus [16]. Long-term potentiation was recorded in the spinal cord in 1993 and recognized as a component of central sensitization [17]. In this setting, stimulation (described as conditioning stimuli) leads to facilitation of neurons that fire autonomously or with subthreshold activation [18,19,20]. Accumulating evidence clearly indicates that a peripheral noxious stimulus is not sufficient to explain all aspects of the pain experience. In addition, there is further corroboration for propagation and modulation of central sensitization at the level of astrocytes, microglia, gap junctions, membrane excitability and gene transcription [21,22,23,24,25,26,27,28,29,30]. 

The major manifestations of central sensitization are pain hypersensitivity characterized by touch allodynia, pressure hyperalgesia, post stimulus sensations (e.g., altered auditory and olfactory) and enhanced temporal summation [10]. Sensitization when provoked in human subjects by stimulation of skin, muscles or viscera can lead to pain hypersensitivity as well as enhanced brain activity detected by electrophysiology or imaging techniques. Central sensitization has been identified as being an important contributor to the pain phenotype in a host of clinical conditions including perceived neuropathic disorders such as fibromyalgia, visceral pain disorders like endometriosis as well as other clinical disorders that present in the absence of identifiable inflammation or neural lesions (e.g., Post Traumatic Stress Disorder (PTSD)). There is no clear consensus on all of the conditions that comprise the class of central sensitivity syndromes. Therefore, we include in this review, a number of conditions that lead to an experience of chronic widespread pain or have been implicated as having central sensitization as a prominent feature in their manifestations [9,10,31]. Although initially thought of as due to direct insult to the CNS such as following a brain or spinal cord lesion, it is now clear that centralized pain can be a component of any peripheral pain disorder [10]. Since about 40 percent of patients appearing in chronic pain clinics carry a diagnosis of fibromyalgia or other central sensitization disorder, the development of rapid and accurate diagnostic capabilities for these conditions is crucial [32,33]. Furthermore, individuals with these disorders represent a large proportion of the population at risk of opiate abuse due to chronic pain despite findings that opioid use corresponded to decreased outcomes in CSS patients [33]. 

Akin to the lack of clear diagnostic markers for central sensitivity disorders is a significant shortage of reliable therapeutic strategies [10]. Patients with poorly explained symptoms are often lumped into the CSS category inappropriately. In fact, most patients who received a diagnosis of CSS from a health professional do not satisfy published criteria [37]. Metabolomics is a rapidly progressing, promising technology that has proven to have diagnostic value and provides unique insight to disease pathology through metabolic pathway analysis. Metabolomics combined with chemometrics allows for non-targeted screening allowing profiling of complex matrices for unknowns or unidentified compounds. In that regard, this form of “omics” provides a powerful tool for identifying currently unknown biomarkers in the area of central sensitivity syndromes. Use of a rapid reproducible biomarker can reassure patients that their symptoms have an objective marker and guide practitioners to direct therapy using non-opioid regimens, since opioids may worsen the pain syndrome and make them more prone to opiate abuse. Technology that can identify key moieties in these disorders may contribute to the identification of novel therapeutic targets and more precise treatment options. As such, the field of metabolomics has substantial potential in improving our understanding of CSS, developing diagnostic tools for these disorders, and developing more effective treatment options for patients suffering from these disorders. All of these factors contribute to the potential that metabolomics has to offer for future therapeutics in CSS. Therefore, it is essential that reliable, accurate biomarkers are identified for these diseases. 

In this review, we describe the current state of metabolomics as it relates to conditions collectively known as central sensitivity syndromes. Because of the complexity of the syndromes noted, our commentary is limited to published data in the field of metabolomics. We acknowledge but do not dwell on biomarker discovery efforts in related fields such as lipidomics, proteomics, immunological discoveries, etc. Furthermore, we recognize that in several of the discussed conditions, the studies presented are either older or metabolomics studies have not been conducted. This should not be construed as if metabolomics is not a fruitful effort in these areas. We take the opposing view that paucity of current research in conditions should represent opportunities for future investigations using the full array of metabolomics tools.

## 2. Current State of Metabolomics Research in CSS

### 2.1. Chronic Fatigue Syndrome (CFS) 

CFS has been associated with errors in the immune system, central nervous system, autonomic nervous system, and energy metabolism, however, these findings are not consistent enough to provide satisfactory sensitivity and/or specificity for a diagnostic test for CFS. As such, the cause of CFS remains unknown [38]. The condition has long been recognized, and has been described by many different terms (e.g., Da Costa's syndrome, effort syndrome, soldier's heart, neurasthenia, myalgic encephalitis /encephalomyelitis, Iceland disease, Akureyri disease, Royal Free disease, immune dysfunction syndrome) [39,40], and most recently, systemic exertion intolerance disease [21]. Clinical characteristics include at least six months of moderate to severe intensity symptoms at least 50% of the time chiefly dominated by fatigue. Criteria include persistent fatigue, post-exertional malaise, sleep problems, cognitive impairment, and orthostatic related symptoms amongst others [41]. 

Recent metabolomics analysis of plasma from CFS subjects and healthy controls revealed significant variations of intermediate metabolites from the tricarboxylic acid and urea cycles [42]. These findings allowed CFS patients to be discriminated from healthy controls based on variances in the ornithine/citrulline and pyruvate/isocitrate ratios (P  <  0.0001 and P   =   0.0069). Further plasma metabolic profiling of CFS subjects and healthy controls quantified 832 metabolites, which identified significantly different concentrations of 14 metabolites related to redox imbalances in CFS subjects [43]. More recently, 361 plasma metabolites were compared between CFS patients and matched controls [44]. 74 of these metabolites were originally found to be significantly altered (P < 0.05), but only 35 were significant after statistical correction. The 35 metabolites were associated with various biological pathways, including taurine metabolism, glyoxylate and dicarboxylate metabolism, purine and amino acid metabolism, and energy metabolism. Another analysis of plasma metabolites assessed 612 metabolites from 63 various biochemical pathways in CFS patients and controls using hydrophilic interaction liquid chromatography, electrospray ionization (ESI), and tandem mass spectrometry (MS/MS) [45,46]. The majority of these metabolite concentrations were decreased in CFS patients compared to controls, but significant abnormalities were identified in 20 unique metabolic pathways. These include an increase in pyrroline-5-carboxylate and decreases in sphingolipids, phospholipids, purines, cholesterol, and microbiome amino acids, to name a few. This finding is consistent with CFS as a hypometabolic state, but the pathways linked to these abnormal metabolite concentrations vary. One pathway consistently affected in CFS subjects is mitochondrial metabolism. Since abnormal mitochondrial function is associated with fatigue, a recent study sequenced mitochondrial DNA (mtDNA) of 93 CFS patients and evaluated the sequences for mutations [47]. No clinically proven mtDNA mutations were realized in the CFS patients, suggesting that mtDNA mutations are not a common element in the etiology of CFS. However, a more recent comparison of mtDNA mutations in CFS subjects and controls identified a significant difference in the percentage of CFS patients lacking a deleterious variant of mtDNA compared to the control group [48]. These findings are interesting, but the importance of these findings on the etiology and pathogenesis of CFS necessitates further study.

A 2003 systematic review analyzed 58 articles that were both directly concerned with CFS and had sample sizes larger than 10. After sorting and reviewing the articles according to quality of methodology, no consistent pattern of immunological abnormalities was identified for CFS patients [49]. 

Urine samples from CFS children and controls were collected before and after exercise to identify possible variances in metabolic profiles between the groups [50]. Pre and post-exercise metabolites were assessed using liquid chromatography-mass spectrometry (LC-MS), Principal Component Analysis (PCA), and orthogonal projections to latent structures-discriminant analysis. Prior to exercise, the CFS group was found to have significantly decreased creatine, indole-acetaldehyde, phytosphingosine, and pyroglutamic acid compared to the control group. Eleven metabolites (nonanedioic acid, methyladenosine, acetylcarnitine, capric acid, corticosterone, creatine, levonorgestrel, pantothenic acid, pyroglutamic acid, xanthosine, and xanthurenic acid) were selected for analysis after exercise. The CFS group demonstrated significantly increased methyladenosine and creatine (P < 0.05), while the other metabolites were significantly decreased (P < 0.05) compared to the control group. The MetPA database was then used to associate the metabolites to their respective metabolic pathways. Prior to exercise, the CFS group exhibited a disordered arginine-proline metabolism pathway. Furthermore, three disordered metabolic pathways (marker metabolites) were identified in the CFS group post-exercise, which were arginine-proline metabolism (creatine), biosynthesis of pantothenic acid and CoA (pantothenic acid), and steroid hormone biosynthesis (corticosterone).

Analysis of peripheral blood mononuclear cells (PBMCs) from 52 CFS patients and 35 healthy controls to assess for consistent cellular patterns of oxidative phosphorylation (OXPHOS) and glycolysis of CFS patients. The data revealed consistently decreased markers of OXPHOS in PBMCs taken from CFS patients compared with healthy controls [51,52]. These markers result from various pathways, such as ATP production and non-mitochondrial respiration. Of note, the markers that best differentiated CFS patients from controls are associated with maximal respiration. Maximal respiration was consistently impaired in CFS patients (P ≤ 0.003). This lower maximal respiration implies that PBMCs in CFS patients cannot respond to extracellular stressors as effectively as healthy PBMCs and, therefore, cannot meet cellular energy needs.

Recently, a reproducible, broad-spectrum metabolomics technique was developed that allows for the simultaneous analysis of over 600 plasma metabolites from many biochemical pathways via a single injection [53]. This technique shows promise as a diagnostic tool for CFS and other multifaceted diseases. 

Ventricular cerebrospinal lactate levels were compared in patients with CFS patients, Fibromyalgia (FM) patients, patients with overlapping CFS/FM, and controls using proton nuclear magnetic resonance spectroscopy (1HNMR) [54]. Although the mean lactate levels did not significantly differ between diseased groups, all three groups had markedly higher lactate levels compared to the healthy volunteers. As such, lactate levels in cerebrospinal fluid (CSF) cannot be used to distinguish FM patients from CFS patients. Contrarily, the increased lactate concentration in patients with central sensitivity syndromes requires further study to determine what role this metabolite has in the various disease states. 

Analysis of fecal and plasma samples from 24 CFS/IBS patients, 26 CFS patients, 1 IBS patient, and 49 controls identified decreased levels of choline, carnitine, and phosphatidylcholine in CFS patients. Additionally, increased levels of phosphatidylethanolmines, triglycerides, and ceramides were identified in CFS patients with IBS comorbidity [55]. Combined analysis of fecal metagenomic and plasma metabolomic data provides a more reliable predictive model for CFS than either data sets alone.

### 2.2. Complex Regional Pain Syndrome (CRPS)

CRPS, formerly known as reflex sympathetic dystrophy, was divided into two types and a new diagnostic criterion was developed after it was determined that the syndrome does not necessarily involve abnormal sympathetic behavior [56,57]. CRPS type I is a syndrome that develops (absent other reasons for these symptoms) after an initiating event, such as a surgery or injury. This condition presents with spontaneous pain, allodynia, and/or hyperalgesia. Edema, blood flow abnormalities, or abnormal sudomotor activity in the painful region may also occur. CRPS type II may develop following a nerve injury. Clinical features include CRPS type I symptoms combined with swelling, skin changes, and/or bone demineralization. CRPS type II frequently manifests after a fracture, soft tissue injury, or surgery [58,59].

Using ^1^H-NMR and multivariate modeling, the metabolic profiles of 105 CSF samples were analyzed from patients with CRPS and healthy volunteers [60]. Significantly different metabolic profiles were identified in the CSF of CRPS patients compared to the volunteers. The specific metabolites that were significantly increased in CRPS subjects were 2-ketoisovalerate, glucose, glutamine, and lactate, while urea concentration was significantly decreased. These findings suggest that chronic CRPS is associated with a catabolic state, likely associated with inflammation. Similarly, urine from both CRPS patients and healthy controls was analyzed using CE–Time of Flight/Mass Spectrometry (TOF/MS) [61]. CRPS samples could be successfully distinguished from healthy samples based on significant differences in several metabolites associated with amplified muscle catabolism, such as increased 2-aminobenzoic acid and decreased creatine. Increased muscle catabolism could further agitate CRPS symptoms and decrease health outcomes. 

### 2.3. Endometriosis 

This term describes a non-cancerous, hormone-dependent, inflammatory disease that can affect females of all ages and hormonal stages, although it typically presents in reproductive years. It involves the development of endometrial tissue outside the uterus. This ectopic endometrial tissue typically develops in the pelvis, but can also appear in the bowel, bladder, and other anatomical locations. The presence of this tissue in abnormal areas can cause inflammation, dysmenorrhea, chronic pain, infertility, and dyspareunia [62,63,64]. 

Recently, the metabolic composition of follicular fluid (FF) was analyzed in women undergoing fertility treatment for tubal diseases (n = 10), unexplained causes of infertility (n = 13), male infertility (n = 10), endometriosis (n = 8), or PCOS (n = 12) was analyzed using NMR [65]. The FF profiles of male infertility, tubal disease, and unexplained cause subjects were not distinguishable by metabolic profiles. Contrarily, the FF profiles of subjects with endometriosis and PCOS could be reliable differentiated from the FF of male infertility subjects due to significantly altered levels of metabolites associated with infertility. A second study compared the metabolic profiles of FF from 17 endometriosis subjects to FF from 16 unaffected subjects via sequential window acquisition of all theoretical fragment-ion spectra (SWATHTM) [66]. In the subjects with endometriosis, significantly different concentrations of phytosphingosine and two lysophosphatidylcholine isomers were identified. These metabolites may contribute to infertility in patients with endometriosis due to their role in essential cellular processes. Another study compared FF metabolites collected from 12 women with ovarian endometriosis to FF from 12 matched controls using NMR [67]. Analysis revealed that endometriosis subjects could be consistently distinguished from controls due to significantly elevated levels of lactate, β-glucose, pyruvate, and valine in the FF. A fourth study utilized 1HNMR and biochemical analyses to develop a metabolic profile of FF from women with stage I-II endometriosis (n = 8), stage III-IV endometriosis (n = 8), and unaffected controls (n = 7) using [68]. Metabolite concentrations from glycolysis, lipid, and amino acid pathways were found to be significantly altered in both endometriosis groups compared to the control group. 

MS/MS was recently used to assess metabolic variation in FF and endometrial tissue from women with stage III-IV endometriosis (n = 10) and healthy controls (n = 10) [69]. Significantly increased concentrations of sphingolipids and phosphatidylcholines were found in both the FF and the endometrial tissue of endometriosis patients compared to control samples. The variation between the metabolomic profile of endometrial tissue located in normal anatomical positions of 29 women with minimal to mild endometriosis patients was compared to similar tissue in 37 healthy women using ultra-high-performance liquid chromatography (UHPLC), high resolution mass spectrometry (HRMS), and ESI [70,71]. Statistical analyses of the differentiating metabolites revealed that levels of uric acid, hypoxanthine, and lysophosphatidylethanolamine could be used to diagnose minimal to mild endometriosis with 66.7% sensitivity and 90.0% specificity. Further analysis of eutopic endometrial tissue via ^1^H-NMR revealed differences in the concentration of metabolites associated with energy, ketogenesis, and gluconeogenesis consistent with various stages of endometriosis [72]. Alanine concentration was found to diagnose Stage 1 endometriosis with 90% sensitivity and 58% specificity. Construction of a regression model of the serum concentrations of alanine, leucine, lysine, proline, and phenylalanine allowed for the diagnosis of Stage II endometriosis with 100% sensitivity and 83% specificity. A third study compared the metabolic profile of eutopic endometrial tissue between 21 patients with either stage I or stage II endometriosis with eutopic endometrial tissue from 20 unaffected women using UHPLC-ESI-HRMS [73]. Analyses revealed significantly decreased concentrations of various phosphatidylcholine, (18:1/22:6), (20:1/14:1), (20:3/20:4), and phosphatidyl-serine (20:3/23:1). Analyses also revealed increased levels of phosphatidic acid (25:5/22:6). Using these metabolites, early endometriosis could be predicted with 90.5% sensitivity and 75.0% specificity. 

Metabolic profiles of peripheral blood samples from laparoscopically confirmed endometriosis subjects (n = 25) and controls (n = 19) were generated using UPLC-ESI-Quadrupole (Q)-TOF techniques [74]. Various lipid, fatty acid, signaling, and cholesterol metabolites were investigated; however, a group of acylcarnitines was identified that consistently and objectively distinguished endometriosis subjects from controls. Unfortunately, the curve analysis outcome was insignificant due to data limitations.

The metabolic profiles of plasma from early-stage endometriosis subjects (n = 6), advanced stage endometriosis subjects (n = 44), and healthy controls (n = 23) were analyzed using ^1^H-NMR and statistical analyses [75]. Metabolic profiles differed significantly between endometriosis patients and controls with subjects from both endometriosis groups having altered metabolites relating to creatine metabolism, branched chain amino acids (BCAA) metabolism, cellular needs, and the syntheses of cell membranes. A second quantification of plasma metabolites in endometriosis patients and controls was conducted using ESI-MS/MS [76]. Over 140 targeted analytes were detected using this technique, including glycerophospholipids, sphingolipids, and acylcarnitines. Eight lipid metabolites were identified at significantly elevated levels in the endometriosis group. A BMI/age-adjusted model was developed using the identified metabolites that allowed for differentiation of endometriosis subjects from control subjects with 90.0% sensitivity and 84.3% specificity. 

Serum from women with stage II-III endometriosis (n = 31) and healthy women (n = 15) was analyzed for predictive metabolites using 1HNMR and pattern recognition. Following metabolite detection, computational methods were utilized to develop a diagnostic model [77]. Results showed significantly altered metabolites from steroid hormone biosynthesis, primary bile acid biosynthesis, biotin metabolism, and taurine/hypotaurine metabolisms in endometriosis subjects compared to control subjects. Positive and negative predictive values for these groups were 71% and 78%, respectively. A second ^1^H-NMR metabolic analysis of serum from laparoscopically confirmed endometriosis subjects (n = 75) compared to healthy controls (n = 60) was performed [78]. Analyses revealed significantly altered concentrations of metabolites associated with glucose metabolism, the Krebs cycle, and mitochondrial deficiencies. These metabolites allowed for classification of endometriosis patients with 92.83% sensitivity and 100% specificity. Further ^1^H-NMR analysis of serum from women with stage I-II endometriosis (n = 22) and age/BMI matched healthy women (n = 23) demonstrated altered metabolite concentrations that correspond to the immune response to endometriosis, increased aerobic glycolysis, faulty pyruvate metabolism, and altered amino acid composition [79]. These metabolite variances allowed endometriosis patients to be discriminated from controls with a sensitivity of 81.8% and a specificity of 91.3%. In a recent case-control study, the serum metabolomic profile was developed in women with stage III-IV endometriosis (n = 50) and controls (n = 50) using MS to test the diagnostic capabilities of serum metabolomics [80]. Using this profile, 10 metabolites were identified based on their significance to the predictive model. The metabolites allowed endometriosis subjects to be distinguished from controls with 84% accuracy.

Peritoneal fluid (PF) from women with endometriosis of the ovaries (n = 29) was compared to PF from healthy women (n = 36) and assessed for invasive metabolites [81]. Of the 148 metabolites quantified, 10 metabolites were found to have significantly decreased concentrations and 125 metabolite ratios were found to be altered in the endometriosis group. The affected metabolites include various forms of carnitine, acylcarnitines, sphingomyelins, and phosphatidylcholines. After adjusting for age, two of the altered metabolite ratios could differentiate endometriosis subjects from controls with 82.8% sensitivity, 94.4% specificity, and an area under the curve (AUC) of 0.944. 

^1^H-NMR analysis of metabolites in the urine of women with early-stage endometriosis (n = 6), advanced stage endometriosis (n = 39), and healthy controls (n = 36) was performed [82]. After statistical analyses, women with endometriosis had consistently altered concentrations of metabolites associated with protein catabolism, inflammatory processes, oxidative stress, and immune response, compared to healthy women. 

### 2.4. Fibromyalgia (FM) 

FM is a significant member of a class of disorders termed central sensitivity syndromes and overlapping chronic pain conditions [83,84,85,86,87]. These conditions have shown significant diagnostic and therapeutic challenges to date. Due to the lack of diagnostic methods for FM, the condition goes undiagnosed in up to 75% of individuals resulting in postponed care [84]. Similarly, the lack of effective treatment options results in substandard care and poor management of patient symptoms. Research suggests that the syndrome arises due to aberrant neuroendocrine function, increased sensitivity to the body’s threat response, and/or dysfunction in the body’s pain receptors [88]. FM presents as general pain and tenderness in consistent locations throughout the body, sleep disturbances relating to this tenderness, and assorted comorbidities [84,85,89,90,91,92]. Fibromyalgia disproportionately affects women and is the leading cause of chronic widespread pain in the United States [85,89,90]. Until a reliable biomarker can be identified, the diagnosis of FM will require in-depth clinical evaluation. The difficult and ambiguous identification of FM persists despite the ability to eliminate other potential causes of pervasive pain through laboratory testing, such as vitamin D deficiency and hypothyroidism. 

A persistent effort in FM research is the search for unambiguous and quantifiable biomarkers that may allow for the identification of at-risk individuals, assist in FM diagnosis, and facilitate more dependable and beneficial treatment of FM patients. Previous studies have shown that metabolomics are especially useful in the study of the brain due to the key role of metabolites in neurological signaling. Similarly, recent studies have reported on the metabolomic disruptions in the brains of fibromyalgia patients [93]. These studies suggest that discrepancies in the metabolic profile of the brain may contribute to neurological syndromes. Recently, the importance of subcategorizing FM patients according to symptoms was realized [94,95]. These findings demonstrate that FM presents in distinct forms which future studies may distinguish as biochemically distinct. Furthermore, the differentiation of FM types may allow for improved evaluation and care for affected patients. 

The comprehensive metabolite profile of urine from subjects with FM (n = 18), first generation family members of the FM subjects (n = 11), age-matched healthy controls (n = 10), and healthy 18–22 year-old subjects (n = 20) was recently analyzed using NMR [96]. The analyses revealed increased concentrations of metabolites related to the gut microbiome, which allowed for successful differentiation of FM individuals from all three control groups with an AUC of 90%. Further NMR analysis of the urine of fibromyalgia patients and matched controls found significantly altered metabolites in the urine of fibromyalgia patients suggesting significant muscle damage compared to controls [36]. Further analysis of urinary metabolites in subjects with persistent FM (n = 18), first generation family members of the FM subjects (n = 11), age-matched healthy controls (n = 10), and healthy 18–22 year-old volunteers (n = 41) [97] via GC-MS was executed. This analysis identified 196 metabolites in the urine; however, the FM subjects could be distinguished from the control groups using 14 significantly increased metabolites that are broadly associated with energy metabolism, carbohydrate degradation and utilization, and host metabolites.

The metabolic profile of plasma from women with FM (n = 22) and healthy women (n = 21) was analyzed using liquid chromatography-Q-TOF/MS) [98]. Many lipid compounds were identified, but the metabolites responsible for discriminating the FM subjects from controls were 1-tetradecanoyl-sn-glycero-3-phosphocholine and 1-hexadecanoyl-sn-glycero-3-phosphocholine. Recent evidence suggests that these metabolites may play a critical role in pain signaling and sensitivity. 

Anti-dense fine speckled 70 (anti-DFS70) antibodies were collected from the blood of FM subjects (n = 39), subjects with systemic lupus erythematosus (SLE) (n = 17), and healthy subjects (n = 19) [99]. These antibodies were subsequently quantified via enzyme-linked immunosorbent assay. The FM and healthy groups had higher levels of anti-DFS70 antibodies compared to the SLE group. Additionally, FM subjects that had either self-endorsed arthralgia or sleep disturbances were found to have significantly higher anti-DFS70 antibody values than those without arthralgia or sleep disturbance.

The first study of metabolomics in FM examined blood from 14 FM patients, 15 osteoarthritis (OA) patients, and 12 rheumatoid arthritis (RA) patients [100] using mid-infrared microspectroscopy (IMS). This analysis allowed for differentiation of RA, OA, and FM patients without error. This finding demonstrates that FM spectral patterns can be consistently separated from other diseases using IMS. Further analysis of group metabolic profiles found that the profiles of the RA and OA groups were similar, while significant differences in metabolites associated with tryptophan catabolism were identified in the FM group. 

More recently, vibrational spectroscopy was performed on blood from various disease groups to differentiate FM patients from RA, OA, and SLE patients (n = 50, n = 29, n = 19, n = 23, respectively) [35]. Distinctive spectral signatures were identified via pattern recognition analysis and patients were organized into disease classes without error. UHPLC, MS/MS, and photodiode array revealed that protein backbones and pyridine-carboxylic acids were significant metabolites in disease discrimination. Furthermore, spectra from FM patients correlated with reported patient reported pain severity. This suggests that these techniques may be capable of differentiating disease severity using biological samples and patient reports. 

### 2.5. Headache 

#### 2.5.1. Tension-Type Headache (TTH) 

TTH is the second-most common disorder worldwide and most diagnosed type of headache [101]. Typically, TTH presents as a non-pulsating, bilateral headache of mild-moderate intensity, without other headache features [102]. In fact, TTH is the least identifiable type of headache due to its, typically, featureless manifestation [103]. Because of this, TTH is not studied as often as other headache disorders.

^1^H-NMR was used to analyze the metabolites present in the basal ganglia (BG) and peritrigonal white matter in SLE subjects with chronic daily headache (CDH) (n = 9), FM subjects with CDH (n = 5), and unaffected controls (n = 6) [104]. Results showed a significantly decreased N-acetylaspartate/creatine ratio in the right BG of SLE patients with CDH compared to both other groups.

#### 2.5.2. Cluster Headache (CH) 

CH is the primary component in a class of headaches known as trigeminal autonomic cephalalgias. These cephalalgias typically present with severe, one-sided pain and symptoms associated with the bodily response to this pain, such as swelling, lacrimation, and facial flushing [82,84]. CH disproportionately affects males and, while less common than TTH, contributes to increased utilization of sick time and work disability [105,106]. Unfortunately, the contributing factors in the etiopathogenesis of CH remain uncertain, leading to decreased health outcomes in these patients [107]. 

A recent systematic review compared metabolic profiles of controls and patients diagnosed with migraine without aura, migraine with aura, CH, CH in remission, and active CH [108]. From the available studies, CH patients displayed significantly altered concentrations of many metabolites, including tyrosine decarboxylase, tyrosine hydroxylase, and levodopa. These findings support metabolomics’ utility in developing the understanding of CH and providing more targeted treatment for CH. 

### 2.6. Idiopathic Low Back Pain 

Recent studies have suggested that up to 80% of individuals will experience some level of low back pain in their lifetime [109,110]. Furthermore, individuals who have back pain for longer than four weeks are more likely to develop chronic back pain that persists for greater than 12 weeks. Occasionally, back pain may result from a serious medical condition, but it is more common for back pain to resolve rather quickly without the need for medical intervention [111]. In 2016, back symptoms were the primary diagnosis for over 1.9% of physician visits in the United States [112]. 

The metabolic profile of plasma was analyzed in patients with chronic low back pain (CLBP) (n = 1128) and healthy controls (n = 760) via hydrophilic interaction ultra-performance liquid chromatography [113]. Significantly increased concentrations of tri-branched and tetra-branched N-glycan arrangements on the glycoproteins of CLBP patients compared to glycoproteins of controls. Additionally, significantly altered concentrations of other glycan structures were identified in CLBP subjects compared to controls, suggesting that the alterations in N-glycosylation may contribute to significant changes in the N-glycome. 

### 2.7. Painful Bladder Syndrome (PBS)/Chronic Prostatitis (CP)/Interstitial Cystitis (IC) 

These are common chronic pain syndromes that have poorly understood pathophysiology [114], but it is likely that the respective symptoms of these diseases result from the contribution of multiple factors; however, these factors are difficult to isolate because these disorders are often accompanied by other chronic pain conditions, such as FM. Recent epidemiological research suggests that 25–30 million women in the US alone may suffer from Bladder Pain Syndrome (BPS)/IC [115] Another epidemiological study of US women found that around 3-6% of women meet diagnostic criteria for BPS/IC, but only about 10% of these women have been diagnosed with BPS/IC [116]. In addition to the persistent pain, dysuria, and nocturnal disturbances reported by individuals with IC, relationships and work-life may be significantly strained. Furthermore, there are substantial financial impacts from IC, such as frequent physician visits, medications, and decreased productivity. Recent studies found discovered that the average yearly cost of IC ranges from $3,631 to $7,043, which is similar or more costly than other chronic pain disorders [117]. 

The understanding of IC has been notably inhibited because reliable, diagnostic tests have not been developed and identified. The discovery of a consistent and specific reliable biomarker would modernize IC diagnosis, allow for more accurate prognosis, and enhance treatment capabilities. This finding would also serve as a novel contribution to the increasing literature addressing the etiology and pathogenesis of IC. Studies have identified possible markers in the urine, bladder tissue, and serum of IC patients [118]. For example, significantly altered levels of antiproliferative factors and growth factors have been identified in the urine of IC patients compared to the urine of healthy controls that show promise as a diagnostic marker [119]. Unfortunately, tests for these biomarkers are not currently available in clinical settings. 

A 2012 review of metabolomic, genomic, proteomic, and other ‘omics’ techniques for IC highlighted a substantial lack of research on the use of ‘omics’ techniques to better understand IC’s pathogenesis, to aid in diagnosing and quantifying the severity of disease, and to allow for the development of more effective treatments for IC [120]. A more recent review on the use of metabolomics in IC supports the aforementioned finding that the use of metabolites in the diagnosis, treatment, and research of IC is underutilized and endorses that the lack of biomarkers in IC is impeding the understanding of this disease, etiology, and possible responses to therapies [121]. GC-MS analysis of urine from IC subjects and controls examined over 450 metabolites and 30 compounds were significantly altered in IC subjects [122]. Unfortunately, the majority of these compounds are unknown, which suggests that the understanding, treatment, and diagnosis of IC may be improved by studying these unknown compounds. 

Analysis of urine samples from IC patients (n = 10), bacterial cystitis patients (n = 10), and healthy controls (n = 10) was recently conducted using UPLC-MS-NMR-MS/MS to identify any metabolites that could be used to discriminate IC subjects [123]. Results showed that phenylacetylglutamine (PAGN) was a significantly altered marker in the urine of IC subjects. Further quantitative methods showed altered ratios of PAGN/creatine ratios that could be used to differentiate IC patients from other patients. Further metabolic analysis of urine from IC patients (n = 43) and healthy controls (n = 20) identified 140 metabolites that were significantly different in the urine of IC patients compared to the urine of healthy controls, with a false discovery rate of only 5% [124]. Based on significant correlation, 15 NMR spikes associated with metabolites in IC were identified as the most reliable markers of IC patients. These metabolites were found to be associated with cellular repair after oxidative stress and neuromodulation of pain. A third study utilized MS to assess the global metabolite profile of urine of patients with IC/bladder pain syndrome (BPS) (n = 40) compared to healthy controls (n = 40) [125]. This analysis allowed for the identification of noteworthy metabolites that are capable of distinguishing IC/BPS subjects from controls. Specifically, etiocholan-3α-ol-17-one sulfate (Etio-S) could identify female IC/BPS subjects with 90% specificity and sensitivity. This metabolite is a reduced isomer of testosterone and its concentration was closely associated with patient reported symptoms. Because of this correlation to symptom scores, Etio-S could be used to separate IC/BPS patients based on severity of disease. Analysis of volatile metabolites in the urine from IC subjects (n = 10) and healthy controls (n = 10) was conducted using solid-phase microextraction-GC-TOF-MS [126]. This analysis found that volatile metabolites had significantly reduced concentrations in the urine of IC patients compared to control urine. Furthermore, these metabolites were associated with inflammatory responses, which indicates they may be used to objectively assess disease severity. 

A bioinformatics approach looked at the urine of IC subjects (n = 50), subjects with bacterial cystitis (n = 30), and healthy controls (n = 47) via MS and ^1^H-NMR to assess for metabolic or protein profiles unique to IC patients [127]. Using this approach, subjects could be discriminated into the respective groups with a success rate of 84%. The metabolic profile of urine from an animal model with nonbacterial prostatitis (NBP) (n = 8), a control group without NBP (n = 8), and a group of NBP animals after treatment with limonin (n = 8) was constructed using UHPLC-ESI-TOF, multivariate analysis (MVA), and pathway analysis [128]. Results identified 20 possible biomarkers for NBP and found significantly altered metabolites from amino acid metabolism and glycerophospholipid metabolism in the NBP group that were normal in the group treated with limolin. This suggests that diagnosis and understanding of NBP may be improved via metabolomics. 

The feasibility of using serum to diagnose IC in humans and domestic cats was assessed via infrared microspectroscopy [129]. Data was obtained from healthy cats (n = 11), cats with feline IC (n = 22), healthy humans (n = 19), and humans with IC (n = 25). This data was used to develop a model using SIMCA. This study was able to successfully predict whether the subject was healthy or had IC with 100% sensitivity using this data. 

### 2.8. Irritable Bowel Syndrome (IBS) 

IBS encompasses a chronic group of disorders of the gastrointestinal tract which typically present with persistent abdominal pain and irregular bowel movements. US estimates suggest that 10–15% of adolescents and adults have IBS related symptoms. Furthermore, IBS concerns and issues contribute to a majority of outpatient visits to gastroenterologists and other providers [130,131,132,133,134,135,136]. IBS can present in various subtypes categorized primarily by symptoms [132]. These are constipation-based IBS (IBS-C), diarrhea-based IBS (IBS-D), a mix between constipation and diarrhea IBS, and unspecified categories. Unfortunately, individuals experiencing IBS symptoms rarely seek medical attention [134,135,136]. In fact, epidemiological studies suggest that 40% of individuals who meet diagnostic criteria for IBS have not been formally diagnosed [135]. Despite underdiagnosis, IBS contributes significantly to higher health care costs and diminished quality of life [136,137,138,139,140]. 

A review of the metabolites as a diagnostic marker in gastrointestinal disorders found promise in volatile organic metabolites (VOMs) [141]. Analysis of VOMs is reproducible and can be performed on patients of all ages and health statuses. These findings support the use of VOMs in the diagnosis and monitoring of gastrointestinal disorders; however, further advancements are needed to improve the accessibility of this diagnostic technique. VOMs were also analyzed in breath samples from IBS patients (n = 170), healthy controls in the clinic setting (n = 153), and controls from the general population (n = 1307) to assess their ability to differentiate IBS patients [142]. Results found that 16 VOMs could accurately predict IBS patients 89.4% of the time and healthy controls 73.3% of the time.

^1^H-NMR was recently used to assess for notable metabolic changes in serum, urine, and stool samples collected from IBS-D patients (n = 8) and healthy controls (n = 16) both before and after intervention with yogurt [143]. Before yogurt consumption, significantly altered concentrations were identified in metabolites from one-carbon metabolism pathways. Notably, results after yogurt consumption indicate the metabolites returned to normal levels after yogurt supplementation. This suggests that metabolomics could allow for better understanding of IBS-D, improved diagnostic capacity, and that further research into symbiotic interventions may allow for improved treatment options. Further analysis of stool metabolites in mice with IBS (n = 6), IBS mice treated with *C. butyricum* (IBS+) (n = 6), and healthy control mice (n = 6) was performed using GC-MS-MVA [144]. The metabolic profiles of the IBS group and control group became progressively different over the two weeks that samples were collected, with 14 significantly altered metabolites in the IBS group. Phenylethylamine was a notable metabolite that correlated with stress in both groups of IBS subjects. Additionally, significant variations in phenylalanine metabolism were identified in the IBS groups. Metabolites associated with steroid hormone biosynthesis, coenzyme A biosynthesis, and the pantothenate biosynthesis pathways were found in different concentrations between the IBS and IBS+ groups. 

The concentration of tryptophan in IBS patients (n = 38) and healthy controls (n = 21) was compared using LC-MS [145]. After IBS patients were further divided into IBS-C and IBS-D groups, results showed significantly lower melatonin/tryptophan levels in the IBS-D group compared to the IBS-C group and healthy controls. Furthermore, kynurenine/tryptophan ratios were lower in both IBS groups compared to healthy controls. 

### 2.9. Migraine 

Migraine is a common episodic disorder characterized by a disabling headache and nausea, light and/or sound sensitivity. These symptoms often occur on a recurrent basis and migraines may escalate in severity over several hours to days. There are four typical phases: the prodrome, the aura, the headache, and the postdrome [146].

Researchers have worked to identify biomarkers in migraine pathology since the 1960s, but these efforts were focused on serotonin metabolism [147]. This focus resulted from the early finding of 5-hydroxyindoleacetic acid, a metabolite of serotonin, in the urine of migraine sufferers. As a result, the genetic and biochemical studies that followed these earlier studies also focused on serotonin synthesis, binding, and transport. Later studies investigated enzymes, receptors, and intermediate metabolites that may affect how serotonin is produced and utilized. More recently, advancements in imaging technologies, metabolomics, and systems biology are being combined to more effectively study serotonergic biology. In general, individuals affected by migraines have dysfunctional neurotransmitter metabolism that can be detected in various sample types. A 2013 review of metabolomic, genomic, proteomic, and other ‘omics’ techniques in migraine patients noted particular metabolites in the pathophysiology of the disease. Pro-inflammatory metabolites, nociceptor associated metabolites, and neutrophins were especially interesting in migraine patients and warrant further investigation [148].

The serum metabolites of individuals suffering from migraines without aura (n = 20) and healthy controls (n = 20) were analyzed using LC-MS to identify migraine biomarkers [149]. Analysis noted 10 significantly decreased metabolites in migraine patients (serotonin and 9 amino acids). The most prominent pathways associated with these metabolites were tryptophan metabolism, arginine and proline metabolism, and aminoacyl-tRNA biosynthesis. Further analysis found that three amino acid differentiating metabolites may be as good or better as a biomarker for migraines than serotonin, which has historically been the predominant migraine biomarker. 

As previously discussed, one study utilized ^1^H-NMR to analyze the metabolites present in the BG and peritrigonal white matter in SLE subjects with chronic daily headache (CDH) (n = 9), FM subjects with CDH (n = 5), and unaffected controls (n = 6) [104]. Results showed a significantly decreased N-acetylaspartate/creatine ratio in the right BG of SLE patients with CDH compared to both other groups.

^1^H-NMR was also used to identify metabolite markers in the plasma of migraine patients (n = 2800) and healthy controls (n = 7353) in a Dutch population [150]. One hundred and forty six individual metabolites and 79 metabolite ratios were quantified using these results. Migraine patients were noted to have significantly decreased apolipoprotein A1 and a decreased ratio of free cholesterol to total lipids in HDLs. Males in the migraine group were also found to have a decreased level of omega-3 fatty acids compared to females and healthy controls. 

^1^H-NMR assessed metabolic variations in the CSF of migraine with aura subjects (n = 38), migraine without aura subjects (n = 27), hemiplegic migraine subjects (n = 18), and healthy controls (n = 43) [151]. 19 metabolites were identified and quantified from the ^1^H-NMR results. Although hemiplegic migraine subjects could be discriminated from controls based on concentrations of 2-hydroxybutyrate and 2-hydroxyisovalerate, the other migraine groups could not be differentiated from controls based on metabolic profiles. 

Despite all the recent advancements in metabolic profiling, the identification and quantification of small polar compounds remains a challenge due to background interferences and low-detection sensitivity. A recent study compared various detection techniques and realized that a combination of MS and matrix-assisted laser desorption/ionization (MALDI) techniques was especially useful for the study of small polar compounds [152]. This allowed for the quantification of 23 amino metabolites in the brain tissue of a murine model. Results showed that cortical spreading depression causes significant neurotransmitter variations compared to healthy samples. As such, MALDI-MS may be useful in furthering the understanding, diagnostic ability, and treatment options for migraine and other neurological disorders. 

### 2.10. Multiple Chemical Sensitivity (MCS) Syndrome

MCS is an independent syndrome associated with broad, intermittent symptoms occurring after minor exposure to various environmental factors [153]. Patient reported symptoms often affect the central nervous system, respiratory system, or the gastrointestinal system and can be debilitating. MCS patients commonly endorse sensitivity to various, unrelated chemical substances and illness following minor exposure to these chemicals [154,155,156]. Unfortunately, consistent physical findings and laboratory tests for diagnosing MCS do not currently exist [157]. Previous studies have consistently implicated the psychological state in MCS, suggesting a behavioral and psychiatric basis in this illness [158]. As such, research supports that many forms of MCS may be a psychiatric condition, rather than a physical one [159]. MCS patients go to drastic measures to avoid exposure to trigger chemicals due to the debilitating symptoms experienced after exposure. This can be a serious impediment to quality of life in MCS individuals. 

Recent metabolomics research compared samples from MCS patients (n = 9) to healthy controls (n = 9) to identify variations in the metabolite profiles that may contribute to the MCS disease state [160]. While 183 metabolites were higher than the normal detection limit, the most notable differences between MCS subjects and controls were significantly altered concentrations of hexanoic acid, pelargonic acid, and acetylcarnitine in patients with MCS. 

### 2.11. Myofascial Pain Syndrome (MPS)

MPS is a localized pain disorder associated with trigger points in the muscles and/or their fascia [161]. With pressure, these tender points result in the MPS-characteristic pain, sensitivity, and response from the autonomic system [162,163]. MPS is a common cause of pain and has a similar clinical manifestation to other chronic pain disorders, such as FM. Unfortunately, consistent and objective diagnostic criteria for MPS is scarce and estimates are unreliable. Furthermore, a significant portion of MPS data is more broadly associated with musculoskeletal pain. As a result, the MPS impact and etiopathology are severely understudied. A study conducted in 1989 estimate that 30% of general internal medicine visits were due to myofascial pain [164]. 

There were no prior metabolomics studies identified using the search term metabolome or metabolomics, but recent studies suggest that corticospinal excitability may be a promising biomarker in the diagnosis and understanding of MPS [165]. 

### 2.12. Polycystic Ovary Syndrome (PCOS) 

PCOS is the most common hormone disorder and cause of infertility in women across the world, affecting between 6.5–8% of all women [166,167]. This disorder often presents during adolescence due to ovary dysfunction and typically has symptoms of hyper-androgenism, such as excessive hair growth, acne, and alopecia [168].

A 2018 systematic review of various metabolites in the diagnosis and treatment in PCOS identified promising metabolites from multiple metabolic pathways. These findings support the use of metabolomics in PCOS research and reveal that metabolomics may allow for an improved understanding of the disease, its etiopathology, and improved treatment and diagnosis [169]. 

Recently, the serum levels of two metabolites (ghrelin and leptin) were recently analyzed in PCOS subjects (n = 130) and healthy controls (n = 121) to assess for metabolic, hormonal, and biochemical variances between groups and between lean/obese subjects in these groups [170]. No significant differences in ghrelin or leptin concentrations were identified between PCOS subjects and healthy controls; however, both of these metabolites were significantly altered in obese PCOS subjects compared to lean PCOS subjects. 

Another study generated serum metabolic profiles from PCOS patients (n = 145) and controls (n = 687) to assess the impact of testosterone levels and obesity on metabolic variances [171] via NMR association analyses. As expected, PCOS women exhibited more metabolic disturbances than controls and several very low-density lipoproteins were altered in this group. These variances in lipid metabolites persisted after adjusting for obesity measures. High-density lipoproteins, apolipoprotein A1, and albumin concentrations were significantly decreased in obese PCOs subjects compared to controls. Furthermore, testosterone levels inversely correlated to insulin levels in the PCOS group, but not in the control group. 

^1^H-NMR analysis of serum from PCOS subjects (n = 74) and controls (n = 68) identified 8 significantly altered amino acids and four altered metabolites of energy metabolism [172]. Pathway analysis of these metabolic variances via MetPA implicate dysfunction in aminoacyl-tRNA biosynthesis, amino acid biosynthesis, and pyruvate metabolism, among others. Lactate, threonine, proline, acetate, and alanine were the most accurate predictors of PCOS. Overall, these findings demonstrate altered carbohydrate, lipid, and amino acid metabolism in PCOS subjects. Further analysis of the serum metabolic profile was conducted on PCOS women (n = 30) and age/BMI matched controls (n = 30) using GC-LC-MS [173]. The subjects in the PCOS group displayed significantly altered metabolites from multiple disrupted metabolic pathways, such as AA synthesis, steroid hormone metabolism, and lipid metabolism. Specifically, PCOS subjects had significantly increased concentrations of phospholipids, organic acids, hormones, and aromatic AA in serum samples compared to the control group and significantly decreased cholesterol levels. Notably, total cholesterol, phenylalanine, uric acid, and lactic acid were among the metabolites used to differentiate PCOS subjects from the control subjects. Further analysis compared the serum metabolites of PCOS subjects (n = 10) and healthy controls (n = 10) via LC-MS to improve the understanding of this disease, its etiopathology, and treatment options [174]. Six unique differentiating metabolites from lipid, androgen, carnitine, and bile acid metabolism pathways were identified. These differentiating metabolites had crucial and distinct roles in disease development. 

In an interventional study, PCOS patients (n = 15) and healthy controls (n = 15) were treated with myo-inositol, D-chiro-inositol, and glucomannan over a three month period [175]. The concentrations of various metabolites and blood components were measured in the serum before and after treatment. Comparing groups before and after treatment showed significantly altered concentrations of 15 relevant metabolites from 12 metabolic pathways

Recently, PCOS subjects (n = 12) and age/BMI matched controls (n = 10) participated in an eight week exercise program at 60% maximal oxygen consumption [176]. Participants had maximal stimulation of insulin resistance and, subsequent, insulin sensitivity measured before and after exercise. The metabolic profile of both groups was measured at baseline and at stimulated max insulin resistance, both before and after exercise. Results found significantly increased concentrations of 8 amino acids in PCOS subjects compared to the control group prior to exercise. Of note, these amino acids were not significantly different between groups following exercise. 

Previous studies have noted a higher prevalence of decreased insulin sensitivity, altered mitochondrial function, and increased insulin resistance (IR) in subjects with PCOS [177,178,179]. Furthermore, adults with IR and either obesity or diabetes typically exhibit altered amino acids (AA), free fatty acids (FFA), and acylcarnitines (AC) profiles. A recent study sought to assess whether these same metabolites are associated with IR/androgens in adolescent PCOS [180]. Teenage, obese girls with PCOS (n = 15) and without PCOS (n = 6) provided plasma samples during a fasting insulin state and hyperinsulinemic state. LC-MS of plasma samples revealed significantly altered AA, AC, and FFA profiles in the PCOS group in both insulin states compared to the control group. Androgen levels had a negative correlation to AC, but no correlation was identified with FFA or AA. These results reveal that young, obese individuals with PCOS exhibit a unique metabolic profile during fasting and hyperinsulinemia that may be associated with IR and metabolic diseases, such as diabetes mellitus. 

UPLC-Q-TOF-MS analysis of plasma from PCOS patients (n = 49) and controls (n = 50) was performed to identify variances in metabolic profiles between these groups [181]. This technique identified abnormal metabolites from lipid and hormone metabolism pathways, such as testosterone, estradiol, low-density lipoprotein, and apolipoprotein B. Significantly different metabolites allowed for the differentiation of PCOS subjects from controls with 100% sensitivity and 86% specificity. These findings support the role of disordered hormone and lipid pathways in the occurrence of PCOS. 

The urinary metabolic profile was analyzed in PCOS subjects (n = 21) and healthy controls (n = 16) using GC-MS to assess for differentiating metabolites [182]. 35 significantly altered metabolites were identified in the PCOS group compared to the control group. Lactose, stearic acid, palmitic acid, and succinic acid were found to be most useful in the differentiation of PCOS subjects. Furthermore, PCA allowed for the grouping of PCOS subjects and controls into two unique units. Of the differentially expressed metabolites, stearic acid, palmitic acid, benzoglycine, and threonine were the most promising differentiating metabolites. 

A UPLC-MS/MS study recently identified 59 altered metabolite concentrations between PCOS patients (n = 22) and controls (n = 15) [183]. After analysis, six metabolites of steroid hormone biosynthetic pathways were determined to be the most promising differentiating metabolites. While the identification of these differentiating metabolites is encouraging, further research must be performed to elucidate their role in the pathogenesis of PCOS and their value in PCOS diagnosis. 

### 2.13. Primary Dysmenorrhea

Dysmenorrhea or painful menstruation, is a common problem of women in their reproductive years and can adversely affect quality of life. Specifically, this condition can cause personal or work-life related problems, such as work absenteeism. Primary dysmenorrhea occurs during menstruation without discernible disease and presents as a recurrent, cramp-like, pain in the lower abdomen of the patient [184,185,186,187,188,189,190,191,192,193]. Secondary dysmenorrhea presents with similar symptoms of primary dysmenorrhea, but the patient will have an obvious disorder associated with these symptoms, such as endometriosis.

Recent analysis of global urinary metabolic profiles were collected from dysmenorrhea patients (n = 36) and controls (n = 27) during luteal regression via UPLC-Q-TOF-MS [194]. After analysis, 10 metabolites from multiple biochemical pathways, such as steroid hormone biosynthesis, amino acid synthesis, and histidine metabolism, were found to be the most promising differentiating metabolites for dysmenorrhea for PD were identified. The differentiating metabolites included citrulline, sphinganine, progesterone, among others. 

### 2.14. Restless Leg Syndrome – Periodic Limb Movement in Sleep 

Restless leg syndrome is a disorder that typically presents with the persistent urge to move one’s legs but it can also be associated with pain and altered sense of touch in other areas of the body [195]. These symptoms are usually worsened at rest, especially at night, and alleviated by moving the affected limb. RLS can detrimentally affect the sleep of affected individuals due to involuntary, jerking movements known as periodic leg movements of sleep.

A recent review of imaging studies in RLS patients suggest that RLS etiopathology may be associated with detrimental changes in the metabolic and functional profiles of neurotransmitters [196]. Specifically, the studies suggest dysfunction in dopaminergic pathways and possibly serotonergic pathways. Functional magnetic resonance imaging identified consistent activation in sensorimotor and limbic areas of the brain. ^1^H-NMR studies confirmed the activation of limbic system and suggest that a glutamatergic dysfunction may be responsible. Finally, NMR studies suggest significantly reduced iron in multiple brain regions of RLS patients compared to healthy individuals. This finding suggests that iron may be a promising metabolite in the study, diagnosis, and treatment of RLS [197]. 

### 2.15. Temporomandibular Disorders (TMD) 

TMD are a class of disorders that involve the jaw and surrounding area. These disorders are common, but have proven difficult to diagnose and classify. TMD may be a subtype of secondary headache disorder and can significantly affect quality of life [86]. Typically, these disorders present with pain in the jaw and surrounding temporomandibular area, difficulty and/or painful chewing, and an aching pain in or around the ear. Management of TMD centers around relieving pain and ensuring normal function of the jaw. Recent estimates in the US suggest that for every 100 million working adults, TMD contributes to almost 18 million lost workdays a year [198]. Recent epidemiological studies found that only 46% of individuals suffering from TMD seek treatment [199]. As such, the identification of a novel biomarker in TMD may improve the understanding of the disease, increase patient outcomes, and encourage those suffering from TMD to seek treatment. There were no prior metabolomics studies identified using the search term metabolome or metabolomics. 

### 2.16. Vulvodynia

Vulvar pain syndrome entails persistent pain of unknown etiology, particularly focused in the vulvar vestibule, and occurring for longer than three months [200,201,202,203,204]. This syndrome has also been referred to as vulvodynia, vestibulodynia, vulvar vestibulitis, and focal vulvitis. Vulvar pain syndrome can be further divided into two broad sub-groups. These subgroups are vulvar pain associated with a specific disorder and vulvodynia, persistent vulvar pain without known cause. According to a 2019 review, no metabolomics studies had been conducted on this area to date [205].

### 2.17. Post-Traumatic Stress Disorder (PTSD) 

PTSD entails the body’s response to a traumatic event that involves genuine or implied damage to an individual or others close to the individual [206]. PTSD typically presents as intrusive memories, avoidance of the traumatic event and things that remind the victim of the event, deleterious changes to thinking and mood, and enhanced physical or emotional reactions, such as being more irritable or nervous. These symptoms result from the combined reaction of the body, mind, and emotional state to triggers associated with the traumatic event [207]. Estimates suggest that roughly 7% of Americans will experience PTSD at some point in their life, but the biochemical processes that occur in this disease state are understudied and not understood [208]. Furthermore, PTSD may be difficult to diagnose because patients may not feel comfortable discussing the traumatic event or may not realize that their physiological responses are associated to event triggers [209]. Metabolomics may allow for improved understanding and diagnostic ability in PTSD and may identify dysfunctional biological pathways that contribute to the disease state.

The variation in metabolites between mice that displayed PTSD behavior and control mice was analyzed via UHPLC-MS/MS-GC-MS [210]. Four-hundred and ninety-six compounds were identified in the groups; however, PTSD mice showed significantly altered concentrations of compounds associated with the body’s stress response pathways. 

An early 2019 review aggregated the current findings in the fields of metabolomics and glycomics in both mice and humans with PTSD [211]. For our purposes, we focus on metabolomic findings and will not discuss glycomic findings. The review identified a trend of increased metabolites from stress-related pathways, such as inflammatory pathways, autoimmune response, and energy metabolism in PTSD animal models. Among the findings of these studies were altered palmitoylethanolamide, glycerophosphoethanolamine, N-acetylaspartate (NAA) to creatine (Cr) ratios, and linolenate, among others. This review endorses a significant lack of metabolomics studies in PTSD patients and supports additional research into the various metabolites in PTSD and their role in the disease. 

A recent study not discussed in the review analyzed the metabolic variations in the amygdala and anterior cingulate cortex (ACC) of PTSD subjects (n = 78) and age/sex matched controls (n = 71) using ^1^H-NMR and magnetic resonance imaging [212]. Significantly increased concentrations of NAA in the ACC and increased concentrations of Cr and myo-inositol (MI) were identified in the amygdala. Changes in NAA and Cr concentrations and the ratio between these two metabolites in PTSD patients may be consistent with increased emotion processing and would be expected as PTSD subjects process stimuli associated with the trigger event [213]. Further ^1^H-NMR analysis of amygdala metabolite concentrations was performed on 28 pediatric PTSD patients and 24 matched controls that were exposed to trauma, but did not develop PTSD [214]. Significantly increased NAA, MI, total Cr, and total choline were identified in the amygdalae of PTSD subjects compared to the non-PTSD trauma group. Additional studies have similarly noted altered NAA concentrations in the ACC of PTSD patients [215,216]. This finding suggests that metabolic variations do exist between PTSD and non-PTSD subjects; however, further studies are needed to determine the influence of these metabolites in disease etiopathology. 

Fasting plasma metabolite profiles were developed for combat veterans with PTSD (n = 52) and combat veterans who did not develop PTSD (n = 51) using UHPLC-MS/MS-GC-MS [217]. Similar to previously discussed studies, analyses identified significant differences in metabolites associated with the body’s stress responses. These altered stress responses [respective metabolite] are listed here: Urea cycle [decreased arginine], TCA cycle [decreased citrate], purine metabolism [increased hypoxanthine], glycolysis [increased pyruvate and glucose], among others. Statistical analysis ruled out comorbid contributions to these metabolic differences, such as smoking, BMI, or depression. As a whole, these findings implicate altered mitochondrial function in the PTSD state and provide direction for future metabolomics studies in this disease.

## 3. Conclusions

In summary, metabolomics may provide a number of clues in the future towards a better understanding of the CSS however, there is much research that remains to be done. None of the conditions has had a reproducible reliable differentiating metabolite identified for diagnostic purposes. It is important to recognize that studies finding differences between two states (disease versus healthy) do not have found biomarkers, but rather differentiating metabolites. When within the same study, validation work is done we can speak about candidate biomarkers and only when these are validated in an independent study can you call them a real biomarker [218]. Using the search terms “metabolome” or “metabolomics” revealed no prior studies for conditions such as myofacial pain syndrome, temporomandibular disorder and vulvodynia to date. However, we recognize that using these terms may miss earlier studies that predated widespread use of the term metabolomics. We found one metabolomics study each for the conditions low back pain, multiple chemical sensitivity syndrome, dysmenorrhea and restless legs syndrome suggesting that opportunities for discovery exist with all of the aforementioned conditions. Reviewing the analysis of the biofluids suggests that the perceived optimal fluid (such as those in closest proximity with the target organ) does not necessarily provide the most illuminating results. This suggests that with our current level of understanding of these disorders, all readily accessible specimens should be evaluated. In some cases, metabolomics has provided promising discoveries with technologies such as vibrational spectroscopy which has led to future potential for biomarker discover in FM and interstitial cystitis. These technologies may lead to future discovery of molecules involved in the pathophysiology of these conditions. In other cases, metabolomics has not made as much of an impact in the last decade. New emerging powerful techniques such as vibrational spectroscopy coupled with ultra HP MS may provide clues to several candidate markers rather than single markers that would give evidence for new therapeutic targets towards the better treatment of these often chronically overlapping conditions, which are a burden on health care resources worldwide. 

## Figures and Tables

**Figure 1 metabolites-10-00164-f001:**
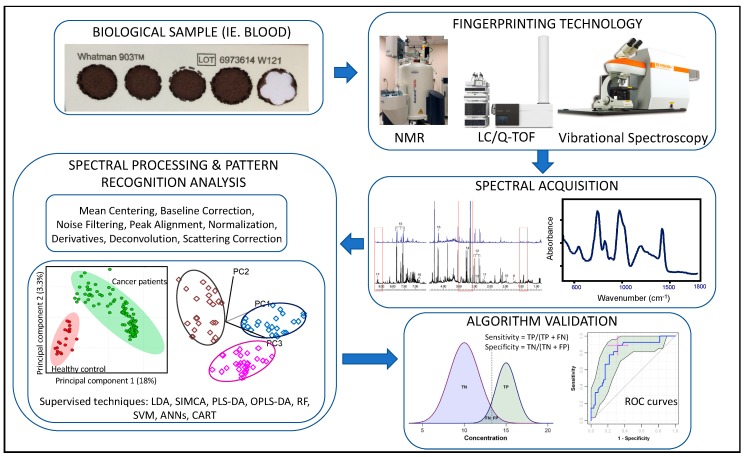
Untargeted metabolomics approach for differentiating metabolite detection [8,34,35,36].

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
