# Peer review of "Metabolomics in Central Sensitivity Syndromes"

_metabolites, 2020, doi:10.3390/metabo10040164_

Round 1

Reviewer 1 Report

This is a comprehensive review of metabolomic  studies in central sensitivity syndromes that comes in a timely manner, as the potential of metabolomics to help uncover disease mechanisms and biomarker candidates has been demonstrated in many human diseases. I have the following suggestions to further improve the paper.

  • What criteria were used to decide which of the disease entities to include ? Is there a consensus definition of CSS that was used to define inclusion criteria?
  • Other than the very general rational for the review (lines 113-117), can the authors provide a mechanistic hypothesis why (of all the advanced profiling technologies) they selected metabolomics? From lines 113-117 any other omics technology would appear to be just as useful.
  • Results: indicate whether the optimal (i.e. in closest proximity with the target organ) biofluids were studied (e.g., study of CSF would appear to be most helpful to study processes in the CNS). This could also be discussed in the Discussion (see 4))
  • The Summary could be much improved by providing a synthesis of findings beyond what is stated in lines 857-862. E.g., is there any type of a common theme? Are there any instructive metabolomic differences or common themes (even if only for subgroups) among the findings on the reviewed disorders?

Author Response

Please see attachments.

We thank the reviewers for their thoughtful critiques regarding our submission. We have amended the manuscript guided by their helpful comments. A summary of the changes to the manuscript responding to their critiques follows:

  1. What criteria were used to decide which of the disease entities to include ? Is there a consensus definition of CSS that was used to define inclusion criteria?

This is addressed in a newly written introduction, lines 98 – 125.

  1. Other than the very general rational for the review (lines 113-117), can the authors provide a mechanistic hypothesis why (of all the advanced profiling technologies) they selected metabolomics? From lines 113-117 any other omics technology would appear to be just as useful.

The following statement has been added to lines 142-145.

Metabolomics combined with chemometrics allows for non-targeted screening allowing profiling of complex matrices for unknowns or unidentified compounds. In that regard, this form of “omics” provides a powerful tool for identifying currently unknown biomarkers in the area of central sensitivity syndromes.

  1. Results: indicate whether the optimal (i.e. in closest proximity with the target organ) biofluids were studied (e.g., study of CSF would appear to be most helpful to study processes in the CNS). This could also be discussed in the Discussion (see 4))

The following statement has been added at line 897 - 901.

  1. The Summary could be much improved by providing a synthesis of findings beyond what is stated in lines 857-862. E.g., is there any type of a common theme? Are there any instructive metabolomic differences or common themes (even if only for subgroups) among the findings on the reviewed disorders?

The summary has been expanded as requested to address the areas mentioned. Lines 891 – 908.

  1. General comments: The authors need to decide for which audience they want to write. Most readers of Metabolites have an understanding of metabolomics, but have little knowledge of CSS. The introduction should therefore focus on CSS and its diagnosis.

We have greatly expanded the discussion of CSS and their hypothesis and diagnosis as noted in lines 98 – 125.

  1. There are quite a few studies discussed that have been published over 10 years ago and still these “discoveries” have not resulted in any new tools or better diagnostic system. This suggest that metabolomics is not a very fruitful approach.

Lines 891-908 have been added to address these concern.

  1. Whilst the paper required a significant amount of work to put together, it was very difficult to read. In a paper such as this I would expect to see a section defining what central sensitization conditions are and some hypothesis as to how the central sensitization induces the pain issues. There are a number of different hypotheses published as to the cause of pain but none of these are presented. These need to be added.

A new paragraph has been included within rewritten lines 98 -125 include the information on central sensitization hypotheses.

Reviewer 2 Report

Comments on metabolites 741988

General comments: The authors need to decide for which audience they want to write. Most readers of Metabolites have an understanding of metabolomics, but have little knowledge of CSS. The introduction should therefore focus on CSS and its diagnosis. The authors need to remember that when the development of biomarkers is completely dependent on the ability to correctly diagnose CSS already. When there is no clear link between the biomarker and the mechanism and the disease is difficult to diagnose, it is not possible to detect nor to validate a biomarker. It would be much more useful when the authors can provide an idea about what would be the best approach to develop and validate biomarkers for CSS and use the literature to explain what useful and less useful approaches are. There are quite a few studies discussed that have been published over 10 years ago and still these “discoveries” have not resulted in any new tools or better diagnostic system. This suggest that metabolomics is not a very fruitful approach.

It would be good when the authors approach this review much more scientifically rather than a stamp collection exercise.

Author Response

(The authors gave the same response as above.)

Reviewer 3 Report

Whilst the paper required a significant amount of work to put together, it was very difficult to read. In a paper such as this I would expect to see a section defining what central sensitization conditions are and some hypothesis as to how the central sensitization induces the pain issues. There are a number of different hypotheses published as to the cause of pain but none of these are presented. These need to be added.

The different sections state studies on various findings and fail to draw conclusions about the findings and their potential role in the development of the issue being discussed. Certain sections describe a claimed CSS condition but provide no metabolome studies. e.g. section O and P. If there are not studies the section should not be added.

Author Response

(The authors gave the same response as above.)

Round 2

Reviewer 2 Report

There is technically not much wrong with this manuscript. My only concern is that it is missed opportunity. When the author would have approached this topic more critically and scientifically this could be a really good paper. 

I will not object any longer to publication, as long as the following changes will be made.

A study finding differences between two states (disease versus healthy) do not have found biomarkers, but differentiating metabolites. When within the same study validation work is done we can speak about candidate biomarkers. And only when this is validated in an independent study you can call this a real biomarker. (see Koulman A, Lane GA, Harrison SJ, Volmer DA. From differentiating metabolites to biomarkers. Anal Bioanal Chem. 2009;394(3):663–670. doi:10.1007/s00216-009-2690-3) So please update the the text. 

So update figure 1 into "untargeted metabolomics approach for differentiating metabolite detection".

Line 336 change "biomarkers" into "differentiating metabolites"

Line 634 change "biomarkers" into "differentiating metabolites"

Line 720 change "biomarkers" into "differentiating metabolites"

Line 724 change "biomarkers" into "differentiating metabolites"

Line 849 change "biomarkers" into "differentiating metabolites"

Line 860 change "biomarkers" into "differentiating metabolites"

Line 897 change "diagnostic biomarkers" into "differentiating metabolites"

Line 906 change "biomarkers" into "differentiating metabolites"

Line 907 change "biomarkers" into "differentiating metabolites"

Line 927/928 change "biomarkers" into "differentiating metabolites"

Line 956 change "biomarker" into "differentiating metabolite"

Author Response

Thank you for the very insightful comments on the review paper.

Note differences in numbering due to format changes.

As requested, Figure 1 has been changed as requested to Untargeted metabolomics approach for differentiating metabolite detection.

Line 336: biomarkers has been changed to differentiating metabolites.

Line 630: biomarkers is changed to differentiating metabolites

Line 735: biomarkers are changed to differentiating metabolites.

Line 737: biomarkers are changed to differentiating metabolites.

Line 773: diagnostic biomarkers is changed to differentiating metabolites

Line 778: biomarkers is changed to differentiating metabolites.

Line 782: biomarkers are changed to differentiating metabolites.

Line 783: biomarkers are changed to differentiating metabolites.

Line 796: biomarkers are changed to differentiating metabolites.

Line 797: biomarkers are changed to differentiating metabolites.

Line 893: biomarker is changed to differentiating metabolite

In addition, added to the summary are the following sentences on line 894 - 897 with corresponding reference [218] as proposed by reviewer.

It is important to recognize that studies finding differences between two disease states (disease versus healthy) do not have found biomarkers, but differentiating metabolites. When within the same study, validation work is done, we can speak about candidate biomarkers. And only when this is validated in an independent study can you call this a real biomarker. 

Reviewer 3 Report

The paper is an improvement but is still very verbose. The paper makes statements that are not true. Example Temporomandibular joint disorders have no metabolome studies to date. There are papers and theses published in the 1990's and after, an example is J Orofac Pain 17(2): 112-124, 2003. These were among the first metabolome studies. No body had invented the word back then so they will not appear in a literature search using metabolome as a search word.

Author Response

Thank you for the insightful comments.

We have modified the text to correct untrue statements as follows:

For all conditions (Myofascial Pain Syndrome, TMD, Vulvodynia) where we stated there were no prior metabolomic studies, we now state:

Line 691: There were no prior metabolomic studies identified using the search term metabolome or metabolomics.

Line 827: There were no prior metabolomic studies identified using the search term metabolome or metabolomics.

Line 836: According to a 2019 review, no metabolomics had been conducted on this area to date.

The summary has been expanded on lines 898 to 901 to reflect the concerns of the reviewer as follows:

Using the search terms metabolome or metabolomics revealed no prior studies for conditions such as Myofascial Pain Syndrome, Temporomandibular disorder and vulvodynia to date. However, we recognize that using these terms may miss earlier studies that predated widespread use of the term metabolomics.

Round 3

Reviewer 3 Report

Verbosity???